# Advances in Molecularly Imprinting Technology for Bioanalytical Applications

**DOI:** 10.3390/s19010177

**Published:** 2019-01-06

**Authors:** Runfa Li, Yonghai Feng, Guoqing Pan, Lei Liu

**Affiliations:** Institute for Advanced Materials, School of Material Science and Engineering, Jiangsu University, Zhenjiang 212013, China; blark_lee@sina.com

**Keywords:** molecular imprinting technology (MIT), molecularly imprinted polymer (MIP), bioprobes, biosensors, biomolecular detection, drug tracking, sensing methods, working mechanisms

## Abstract

In recent years, along with the rapid development of relevant biological fields, there has been a tremendous motivation to combine molecular imprinting technology (MIT) with biosensing. In this situation, bioprobes and biosensors based on molecularly imprinted polymers (MIPs) have emerged as a reliable candidate for a comprehensive range of applications, from biomolecule detection to drug tracking. Unlike their precursors such as classic immunosensors based on antibody binding and natural receptor elements, MIPs create complementary cavities with stronger binding affinity, while their intrinsic artificial polymers facilitate their use in harsh environments. The major objective of this work is to review recent MIP bioprobes and biosensors, especially those used for biomolecules and drugs. In this review, MIP bioprobes and biosensors are categorized by sensing method, including optical sensing, electrochemical sensing, gravimetric sensing and magnetic sensing, respectively. The working mechanism(s) of each sensing method are thoroughly discussed. Moreover, this work aims to present the cutting-edge structures and modifiers offering higher properties and performances, and clearly point out recent efforts dedicated to introduce multi-sensing and multi-functional MIP bioprobes and biosensors applicable to interdisciplinary fields.

## 1. Introduction

As a powerful molecular synthetic strategy, molecular imprinting technology (MIT) has been studied since first presented by Wulff [1] 40 years ago. It has come a long way in recent decades as a technique in a broad range of solid phase extraction, chemical reaction catalysis, membrane separation or chromatographic separation, biosensing and cell recognition applications [2,3,4,5]. The molecular imprinting process is straightforward, typically involving the copolymerization of a functional monomer or a series of functional monomers, a cross-linker, and an initiator in the presence of template molecules, whereby the functional monomer possesses special functional groups to bind with the template molecules. After polymerization, the resulting molecularly imprinted polymers (MIPs) have stabilized binding moieties and subsequent removal of the template molecules leaves a cavity with complementary size, shape and chemical moieties to the template and facilitates the capture of analytes by the MIP [6,7,8,9]. A schematic of the whole process is shown in Figure 1.

A bioprobe or a biosensor is a sensing kit that translates the first-hand sensing signals, for example, optical signals or mass signals, into specific quantitative or semi-quantitative analytical information. It is an integrated device containing two major components: the recognition receptor, which interacts with the target molecule; and the transducer, which is responsible for converting the first-hand signals into measurable information. Most reviews and books utilize the generalized term “biosensors” to define bioprobes and biosensors as one. Although they work for the same purpose, herein, we use separate terms because we aspire to clarify and stress the distinction that MIP bioprobes are fabricated as composite nanoparticles working in dispersed phases, especially in solutions, whereas MIP biosensors are manufactured as layer-by-layer physical constructs.

It is obvious from the literature that researchers are gradually transferring their focus to employing MIPs as alternative bioreceptors to build bioprobes and biosensors rather than using traditional approaches such as antigen-antibody binding or with the help of enzymes, proteins, even whole cells. This is not a coincidence because MIPs have an intrinsic edge over those natural recognition elements. As an artificial polymer nanofilm, a MIP is intrinsically stable and capable of working in extreme environments such as an organic solvent, acid or base, at high temperature and/or high pressure. Besides, MIPs allow more modifiers to work together in a whole system and enable innovative tailor-made bioprobe and biosensor structures with further properties.

The aim of this work is to systemically review recent MIP bioprobes and biosensors applied to biological detection and drug tracking. We organized this review by distinct sensing methods, which include optical sensing, electrochemical sensing, gravimetric sensing and magnetic sensing, presenting thoroughly analyzed mechanisms for each sensing approach. Simultaneously, this review presents multifarious cutting-edge structures and the incorporation of distinct modifiers into MIP bioprobes and biosensors, and compiles multi-sensing and even multi-functional designs with enhanced properties and performance. We believe that this review represents a panoramic sketch as well as a continuous incentive for researchers to further exploit the potential of MIP bioprobes and biosensors.

## 2. Optical Sensing

Optical sensing is a major sensing approach for MIP bioprobes and biosensors, which involves a process of transforming firsthand optical signals into detectable electric signals. Herein, we review optical sensing by three optical detecting mechanisms, namely, the fluorescence quenching mechanism, fluorescence enhancing mechanism, and surface plasmon resonance mechanism. Detailed information about optical sensing with MIP bioprobes and biosensors is summarized in Table 1.

### 2.1. Fluorescence Quenching Mechanism

Optical signals are obtained from quantum dots (QDs) as luminescent bioprobes or luminescent biosensors. QDs are nanoparticles that have superior properties in the field of biolabeling and bioimaging.

Through the process of fluorescence quenching, the change of QDs’ fluorescence intensity can be detected as an optical signal when MIP bioprobes or biosensors rebind with template molecules. Typically, three mechanisms account for the fluorescence quenching of QDs, where template molecules act as quenchers. The major cause of the QDs quenching is photo-induced electron transfer (PET), which occurs when excited QDs capture excitons offered by template molecules (donor) through photo-induced oxidation, or transfer excitons to the template (acceptor) through photo-induced reduction. By capturing or transferring electrons, QDs relax back to ground state and achieve fluorescence quenching. The mechanism is shown in Figure 2A,B.Photo-induced Oxidation:[QDs]^2+^ + hν → [QDs]^2+*^[QDs]^2+*^ + donor → [QDs]^+^ + donor^+^Photo-induced Reduction:[QDs]^2+^ + hν → [QDs]^2+*^[QDs]^2+*^ + acceptor → [QDs]^3+^ + acceptor^−^

The other two mechanisms for fluorescence quenching are Dexter electron transfer (collisional energy transfer) and fluorescence resonance energy transfer (FRET), which can both occur when the emission spectrum of QDs (donor) overlap the absorption spectrum of template molecules (acceptor). The mechanism is shown in Figure 2C.

QDs fluorescence quenching process follows Stern-Volmer relation:$$\frac{F_{0}}{F}=1+K_{SV}C_{q}$$
where *F*_0_ and *F* are the fluorescence intensity in the absence and presence of quencher, respectively, *K_sv_* is the quenching constant (Stern-Volmer constant) for the quencher, and *C_q_* is the concentration of the quencher.

Traditional organic dyes and typical fluorophores are in a trend to be replaced by QDs bioprobes and biosensors due to their superior performance in biolabling and bioimaging. Preponderance of QDs can be generalized as:1)Large Stokes Shifts of QDs avoid fluorescence self-quenching.2)Regularly sized QDs possess the emission spectrum with sharper and narrower peaks than typical fluorophores.3)QDs do not have the long-wavelength tail which interfere with the application of multi-fluorophores imaging or multi-analyte measurements.4)The emission spectrum of the QDs are roughly symmetrical on the wavelength scale.5)Their absorption spectrum at broad wavelength ranging from shorter wavelength than the onset of the absorption spectrum of organic dyes, which means QDs can be excited by a spectrally broad light source and use the same excitation light source to excite various size of QDs, make it accessible for easy muti-labeling as well as simultaneously detecting.6)QDs possess strong photoluminescence with the fluorescent intensity tenfold more than traditional organic dyes.7)The photostability of QDs exhibites 100 times more than traditional organic dyes.

To make the best use of QDs, combining QDs with MIP to fabricate QDs@MIP bioprobes or biosensors has become a research focus. Following the fluorescence quenching mechanism, QDs@MIP biological probes or biosensors are fabricated as various structures in different strategies to maximize the preponderance of QDs in optical signals for biolabeling and biosensing.

In order to functionalize as MIP bioprobes or biosensors, QDs should be fabricated in core-shell structures. High surface area is always an important parameter for nanoparticles, but this property may be problematic for QDs because it leads to reduced luminescence efficiency and photobleaching. However, the core-shell structure can solve the problem. The core-shell structure has high quantum yields and enhanced photochemical stability, and by enclosing the core QDs with another material having a larger bandgap, we can confine the excitation to the core, reducing nonradiative relaxation pathways and avoiding photobleaching. Another reason for fabricating a core-shell structure is that the QD core is always a heavy metal, and the shells can serve as protective shields to prevent toxicity due to leakage, making in-vivo or in-vitro studies possible. Sol-gel polymerization is the traditional and classical method to synthesize MIP bioprobes, which involves conversion of monomers into a colloidal solution (sol) that acts as the precursor for an integrated network (or gel) of either discrete particles or network polymers. In recent years, many advanced synthesis strategies have also appeared, such as ultrasound irradiation combined with precipitation polymerization, which help synthesize doped QDs and MIP-coated composites. This strategy ensures the whole synthesis can be accomplished within 4 h. Another advanced synthesis strategy is micro-emulsion polymerization. The advantages of this strategy can be generalized as involving a thermodynamically stable emulsion, high surfactant concentration, and lower monomer concentration. Table 1 shows detailed information of the synthesis approach and imprinting techniques used to produce MIP bioprobes and biosensors for optical sensing.

Concretely, metal atoms (including heavy metals) are almost always selected to be the core materials, and few studies have reported other core materials like SiO_2_ nanoparticles or polymer nanoparticles. Silica and polysilane are optimum shell materials due to several reasons. First, the band gaps of silica are strictly larger than those of metals, which means a silica shell allows the excitation to be confined to the metal core, thus avoiding photobleaching of QDs. Second, the multivalency of extended polysilane ensures the solubility of QDs. Third, the multivalency of extended polysilane offers various bonding options with polymers, by contrast, the single direct bond of thiols on core metals limits the possibility to bind with various polymers. In a standard polymerization process, 3-aminopropyltriethoxysilane (APTES) and tetraethoxysilane (TEOS) as polymeric monomer and cross-linker, respectively, form a SiO_2_ nanolayer through hydrolysis and condensation reactions, finally, the MIP layer is polymerized at the SiO_2_ nanolayer followed by the elution of template molecules to produce complementary rebinding cavities.

#### 2.1.1. QDs as the Core

Typical heavy metals, for example, CdSe [14], CdTe [15,24,25], InP [13,26], compounds not metals are most widely used as QDs core. Light QD metals such as FeSe [10] are also reported these days. Polymerizing the SiO_2_ layer on the surface of the QD metal core is followed by fluorospectrometry for analysis of sensing and rebinding of the QD bioprobes. Doped metal composites have gradually replaced pure metals as QD cores. Doped QDs maintain all the advantages of QDs but are free of self-quenching due to the large Stokes shift. Adjacent QDs may experience a fluorescence resonance energy transmission process and result in fluorescence quenching, but doped QDs with smaller band gaps avoid the quenching. Doped QDs also have longer lifetimes and lower cytotoxicity. The long lifetime of doped QDs make it possible to produce longer emission times eclipsing the background fluorescence interferences, while the introduction of light metals as dopants offers great opportunities to lower the toxicity of QDs.

To dat, the host QDs are those with large band gaps, especially ZnS (Eg ~ 3.6 eV), and dopants could be CdSe [27], CdTe [28] or Mn [10,11,13,29], among which, Mn doped ZnS QDs has been thoroughly studied and most widely used for producing QDs@MIP bioprobes and biosensors.

Tan fabricated Mn doped ZnS QDs@SiO_2_@MIP bioprobes [30] for the detection of bovine hemoglobin (BHb). The probes, with a diameter of 7 nm, emit a strong orange phosphorescence with a sharp emission peak at 580–590 nm, which is attributed to the ^4^T_1_-^6^A_1_ transitional emission. Fluorospectrometry clearly exhibits the decrease of fluorescence intensity when the probe rebinds with BHb. High selectivity is demonstrated by a high imprinting factor (IF) of 3.1, which is the ratio of *K_SV_* (Stern-Volmer constant) of the MIP- and NIP-probes, indicating that the molecular imprinting process greatly promotes the quenching efficiency of QDs as well as enhances the spectral sensitivity of the MIP probes to the template protein.

In recent years, carbon materials have been introduced as QD bioprobe cores due to their unique advantages. Carbon quantum dots (CQDs) [31,32], fullerene quantum dots (FQDs) [33] and graphene quantum dots (GQDs) [17,34] are the most widely studied examples. QDs made of carbon materials offer unique optical and biological superiority while retaining the good properties of typical semiconductor QDs. First, QDs made of carbon materials have characteristic photoinduced electron transfer (PET) properties, because carbon materials are available both as electron donors and electron acceptors, which enables more efficient fluorescence quenching. Second, QDs made of carbon materials have no cytotoxicity, which makes them good alternatives to the typical semiconductor QDs containing heavy metals. Furthermore, QDs of carbon materials have been demonstrated to have high solubility, robust chemical stability and photostability, in addition to high biocompatibility.

With the increasing requirement of accuracy and multifunction, QDs@MIP bioprobes are no longer limited to the single emission wavelengths of single-QDs materials, and different dual- [12,16] or multi-QDs@MIP bioprobe structures have been successfully fabricated to reduce errors and achieve multiplexed imaging. According to the Stern-Volmer equation, *F*_0_ is the emission fluorescence intensity of bioprobes in the absence of the quencher, however, when measuring *F*_0_, it is inevitable that fluorescence quenching of the bioprobe in fact occurs in the presence of quenchers through a collisional or PET quenching mechanism, inducing inaccurate measurements of *F*_0_ (the reference signal). In order to eliminate the error, Amjadi [16] replaced the initial Stern-Volmer equation with a modified method, the ratiometic method. The group designed a novel core-shell structure of a dual-QDs@MIP bioprobe by enclosing CDs in an inner silica core with CdTe/CdS QDs and MIP cavities surrounded by the outer silica shell, as shown in Figure 3A, CTAB as surfactant produced a mesoporous silica outer layer and enhanced the surface area of the dual-QDs@MIP bioprobe. In this structure, the emission of CDs is stable and is set to be the reference signal, while the outer CdTe/CdS QDs experience collisional or PET fluorescence quenching, serving as the responsive signal label. The fluorescence emission spectrum demonstrates that with the increasing concentration of DNZ (the target analyte), the fluorescence intensity of CdTe/CdS QDs (the response signal) obviously decreases while the fluorescence intensity of CDs (the reference signal) hardly changes. In this situation, the study specifically measured the change of the ratio of QDs’ fluorescent intensity to CDs fluorescent intensity. Since the intensity of CDs is stable, it avoids involving *F*_0_ and the consequent possible error. Panagiotopoulou et al. [12] designed another dual-QDs@MIP bioprobe system by separately using two InP/ZnS QDs of different colors to biolabel human keratinocytes simultaneously as shown in Figure 3B. This multiplexed sensing strategy paves the way for further applications in biomolecular detection as well as clinical diagnosis.

#### 2.1.2. Other Materials as the Core

QDs as the core of MIP bioprobes is the typical structure, but other materials are also eligible for use as cores due to their unique properties. To date, only silica/magnetic silica nanoparticles [11,16,35] and polymeric nanoparticles [26] have been designed to be the core of MIP bioprobes. Jia [18] fabricated a sandwich-like bioprobe structure by putting CdTe QDs in the middle of a silica core and a silica layer; in this structure, one nanoparticle can be treated with plenty of QDs, enhancing the fluorescent intensity. Moreover, CTAB serves as surfactant producing a mesoporous silica layer, with extremely enhanced binding efficiency. Zor [19] synthesized a novel bioprobe structure by using magnetic silica nanobeads as the core covered by graphene QDs, finally polymerizing and producing MIP cavities on the outermost layer as shown in Figure 3C. QDs cannot be directly grafted onto the surface of magnetic nanoparticles, however, in this study, silica was involved to serve as the intermediate nanolayer connecting QDs to the magnetic nanocore. As a consequence, this novel structure combines optical sensing with magnetic properties to simultaneously bioimage the target analyte while capturing and preconcentrating it. Moreover, polymeric nanoparticles are also available to serve as the core. Zhou [26] used polyindole (PIn) as the nanocore as shown in Figure 3D, which possesses attributes including slow degradation rate, good thermal stability, and high redox activity. Based on these advantages, the group successfully sythesized PIn/GQDs@MIPs bioprobes for dopamine.

### 2.2. Fluorescence Enhancing Mechanism

Fluorescence quenching is the major mechanism applied in QDs@MIP bioprobes and biosensors, whereas the fluorescence enhancing mechanism is also desirable. Several groups have reported the successful synthesis of QDs@MIP bioprobes on the basis of the fluorescence enhancing mechanism [36,37,38,39,40,41,42,43,44,45,46]. The Stern-Volmer equation is still applicable for the fluorescence enhancing mechanism, but $C_{q}$ here means the concentration of template molecules instead that of quenchers. The Langmuir binding isothermal equation also can be used to describe the relationship between the fluorescence enhancing and the concentration of template molecules. The equation can be linearized to take the following form:$$\frac{C}{I}=\left(\frac{1}{BI_{\max}}\right)+\left(\frac{1}{I_{\max}}\right)C$$
where *I*_max_ and *I* are the maximum fluorescence intensity and the intensity at a given concentration of template molecules, respectively. Although the fluorescence enhancing mechanism is even more complex than the common fluorescence quenching mechanism, this process can be basically attributed to the FRET system, the PET system, and the surface passivation process of QDs.

#### 2.2.1. PET System

The PET system is the major system of the fluorescence enhancing mechanism [36,37,38,39,40,41]. In a PET process, lone electrons or lone pair electrons of the fluorescent organic functional monomers can experience a PET process to transfer electrons away, and fluorescence enhancement occurs when the rebinding of bioprobes to analytes inhibits PET. The core idea of this method is to deliberately choose suitable fluorescent organic functional monomers. Wang [36] introduced nitrobenzoxadiazole (NBD) as the detection signal source to fabricate QDs@SiO_2_ bioprobes as shown in Figure 4A. The amine group of APTES with nitrogen lone-pair electrons can quench an adjacent NBD by a N-to-NBD PET process. However, the PET process is inhibited when the amine group rebinds with analytes, and an obvious fluorescence enhancement is exhibited. Also using NBD as fluorescent monomers, Wan [37] synthesized “light up” bioprobes with a silica core and a MIP shell exhibiting same efficient fluorescence enhancing. Kubo [41] used 2-acrylamidoquinoline as fluorescent monomer, and when the monomer rebinds with target molecules, the formation of a hydrogen bond to the nitrogen is similar to protonating the nitrogen, inhibiting the PET process of the quinoline nitrogen, and achieving fluorescence enhancing as shown in Figure 4B. Other groups [38,39,40] have also developed different MIP bioprobes by displacing different organic monomers, however, the basis of this method is by inhibiting PET system.

#### 2.2.2. FRET Systems

FRET is process of energy transfer from a donor to an acceptor. Fluorescence enhancement occurs when FRET is inhibited. Gao [43] and Xia [42] both synthesized the same fluorescence enhancing MIP bioprobe structure on the basis of FRET as shown in Figure 4C. In the absence of template molecules, FRET occurs from the QDs (energy donors) to Au NPs (energy acceptors), leading to the energy loss of QDs whose fluorescence intensity stays at a low level. However, in the presence of template molecules, the rebinding process of bioprobes with analytes inhibits the FRET process from QDs to Au NPs, thus enhancing the fluorescence intensities of QDs. The expected result was demonstrated by fluorospectrometry, which shows an enhancement of FL peak with the increasing concentration of analytes.

#### 2.2.3. Surface Passivation System

It is also reported that the fluorescence enhancing mechanism can be attributed to the restoration of QD surface defects, namely through the surface passivation system [45] as shown in Figure 4D.

Wang [44] synthesized CdTe QDs capped with thioglycolic acid (TGA)@MIP bioprobes for the detection of cysteine (Cys). The terminal thiol groups of Cys are easily grafted to the surface of CdTe QDs, inducing plenty of Cys conjugated to CdTe QDs which leads to the decrease of the surface defects of QDs. The surface passivation system has been demonstrated by Wang through fluorescence titration experiments by l-alanine and l-tryptophan, where the resultant fluorescent spectra shows obvious increases of the emission peaks with the increasing concentration of Cys.

### 2.3. Surface Plasmon Resonance Mechanism

It is reported that the surface plasmon resonance (SPR) mechanism has become a widely used strategy to build MIP bioprobes and biosensors. Unlike the fluorescence mechanism, as an optical sensing method, SPR lacks a self-emitting light and instead works by means of detecting the changes of angles of incidence. Plasmons are oscillations of conduction band electrons occurring at the metal-dielectric interface; these oscillations induce an electric field whose intensity declines extremely in a dielectric medium. When an incident beam reaches the surface of a metal at a specific angle, an evanescent field comes into being. If the evanescent field is resonant with surface plasmons, which means energy is transferred from photons to electrons, a surface plasmon resonance occurs, followed by a tremendous decay of the reflected light intensity at a specific incident angle which can be detected by a SPR detector. To excite the surface plasmon over the metal-dielectric interface, an incident beam is typically passed through a gold-coated glass. With the decoration or addition of a film onto the planar metal surface, the dielectric area become thicker and the refractive index changes, which induces the resonance curves shift to a higher angle, and this shift exhibits the assembly process of the SPR MIP biosensors. For the same reason, at a certain incidence angle, SPR signals are enhanced when the MIP biosensors rebind with template molecules. MIP biosensors have a series of advantages over those of QDs such as real-time sensing and direct, label-free detection. However, a classical SPR device involving a gold or silver film to induce a SPR process only enables sensing for macro-biomolecules, the detection limit is not sufficiently low in terms of sensing light weight biomolecules including DNA, or hormones. When rebinding with light biomolecules, the change of refractive index is too small to be observable. As a result, further decorations must be added onto the classical gold or silver films for enhancing their bio-sensibility and accuracy in terms of light biomolecules detection.

#### 2.3.1. Nanoparticle Decorations

Nanoparticles (NPs) decorated on a Au film is the most widely used strategy to enhance SPR signals. Metallic nanoparticles such as Au or Ag NPs have been demonstrated as amplifying labels for SPR@MIP biosensors. Electric coupling occurs to the localized surface plasmons of Au NPs with the surface plasmon waves produced by the Au film, shifting the SPR energy which induces an obvious decay of SPR signals, thus enhancing the SPR signal change [47]. Selecting thioaniline as the functional monomer, Riskin [48,49,50] built a SPR@MIP biosensor in a series of studies for different analytes where molecularly imprinted Au NPs were successfully grafted to a gold-coated glass surface. Based on this structure, the group fabricated a SPR@MIP biosensor for the detection of different amino acids by introducing cysteine as another functional monomer associated with Au NPs to bind with different amino acids through complementary zwitterionic electrical interactions and hydrogen bond interactions. Results from the sensogram indicated that Au NP decoration enhanced the SPR signals greatly.

Besides metal NPs, semiconductor QDs are also involved for the decoration of SPR metal films. It was demonstrated that semiconductor QDs will experience an electric separation process by transferring the conduction band electrons to Au NPs [51,52,53]. In this charging process of Au NPs, because the localized surface plasmon (LSPR) of Au NPs experiences electric coupling with surface plasmon of the gold film along the metal-dielectric interface, not only Au NPs will exhibit LSPR shifts, but also the surface plasmon of the Au film will be altered, inducing changes in SPR signals. Since Zayats [51] first integrated SPR spectroscopy with CdS QDs charging of Au NPs, many groups have managed to introduce QDs into SPR biosensors. Sensorgrams indicate the great properties of these new SPR biosensors including apparent enhancement of SPR signals. However, as far as we know, antibodies have been the only decoration type for this novel QDs-SPR biosensor so far. As a result, MIP decoration of biosensors should be further exploited, and we firmly believe that with MIPs combined with QDs-SPR biosensors, the properties, including SPR bio-sensitivity, will be exploited to a higher level.

#### 2.3.2. Layer Decorations

It has been demonstrated that layer decoration is an even better approach to exploit properties of SPR biosensors compared with NPs decoration. According to reported studies [54,55,56,57], graphene sheets (GO) and graphene oxide sheets (GOS) are two major materials for layer decoration of SPR biosensors. Considering multiple oxygen-containing functional groups on the terminal of GOS which increase the dielectric constant, Chiu [57] decorated GOS onto the typical Au film of a SPR glass prism. Results indicated enhanced properties compared with a typical gold film-coated SPR, including a 5.2 times higher affinity constant, 2 time higher SPR angle shifts and 1/100th lower detection limits. Zeng [55] replaced GOS with pristine graphene due to its higher electron transfer rate and higher optical transparency. With Au NPs decorated on GO, a fourfold sensitivity enhancement was achieved compared to a typical SPR. Because the longitudinal band of Au naonorods (NRs) is very sensitive to causing electron oscillations along the long axis of Au NRs, thus inducing rather stronger SPR signals, Zhang [54] replaced Au NPs with Au NRs managing to achieve a 32-fold lower detection limit. Recently, inspired by the great optical absorption property of MoS_2_ for enhancing charge transfer efficiency, Zeng [56] even enhanced the sensitivity by more than 500 times by decorating a MoS_2_ nanolayer between graphene layer and gold film. Unfortunately, so far all reported studies have been limited to using antigen-antibody sensing, as a result, there is a tremendous urgency to upgrade the old antigen-antibody sensing to the decoration with MIPs, which would certainly enhance the selectivity and other properties of SPR biosensors to a higher level.

#### 2.3.3. New Structures

A series of novel SPR@MIP biosensors based on optical fibers [58,59,60,61,62] has aroused great interest due to its absence of complex electric devices compared to the typical sandwich-like SPR@MIP biosensor. This SPR@MIP-fiber biosensor can be prepared by removing a small part of the cladding and decorating it with a metal film and a MIP film, and then white light is launched at one terminal of the fiber and detected at the other end of the fiber, the decrease of light intensity through the transmission in the fiber being collected as the SPR signal. Although it is reported that this structure does not enhance the selectivity, it enables on-line and remote sensing while reducing the cost and dimensions due to the absence of complex electric devices. Verma [58] used a silver layer to fabricate the biosensor for successfully detecting melamine and vitamin B3, respectively. Cennamo [59,60,61] improved the properties of optical fiber SPR@MIP biosensors by successively replacing the typical optical fibers (POF) with tapered optical fibers and substituting the typical Au film or Au NPs with five-branch gold nanostars (GOS) as shown in Figure 5A. A tapering process led to the modification of the diameter and the core refractive index, causing an enhancement of the SPR signals. GOS enables a higher surface area while presenting multiple resonances which enhance the sensitivity. The results indicated that the sensitivity of this novel structure is about three (vs not-tapered POF) or thirty (vs tapered POF) times that of POF biosensors based on MIPs decorated on a bare gold layer.

Another representative structure of MIP biosensors based on SPR is the hybrid microgel. The motion of microgels is not restricted to a substrate, so they have superior performance in the solvent phase compared with other structures. When exposed to stimuli, microgels are apt to swell or shrink due to rebinding with or separation from analytes (stimuli), inducing a phase volume transition and a change of SPR signals. Wu [63] designed a novel molecularly imprinted glucose-responsive microgel as SPR bioprobe made of Ag NPs and NIPAM-AAm-VPBA as functional monomer as shown in Figure 5B. When binding with glucose, the microgel undergoes a swelling of phase volume, resulting in an expansion of the distance between Ag NPs, hence followed by a SPR signal shift induced by the changing vicinal environment of the Ag NPs. Amazingly, this novel microgel enables one to determine glucose concentration levels without instrumental aid by changing color from yellow to red when the glucose concentration reached 20.0 mM. As demonstrated by real sample analysis, this MIP microgel displays sufficient accuracy comparable to the clinical standard.

Recently, another cutting-edge hollow plasmonic nanostructure has been designed based on gold nanocages (AuNCs) [64] as shown in Figure 5C, exhibiting even higher refractive index sensitivity over that of the AuNPs and AuNRs reviewed above, as a result of highly tunable localized surface plasmon resonance into the near infrared region. Compared to the visible spectral range, the endogenous absorption coefficient of living tissue is nearly two orders of magnitude smaller within this near-infrared area. Furthermore, AuNCs exhibit drastic scattering and absorption cross-sections which are significant properties for photoacoustic imaging and photothermal therapy. Combined with molecular imprinting technology, the AuNCs are polymerized with APTES and TEOS as the sillanization process to fabricate a AuNCs@MIP bioprobe based on SPR. Results showed that the bioprobe achieved more than an order of magnitude lower detection limit for sensing urinary proteins which enables the detection of a kidney injuries down to the concentration of 25 ng mL^−1^.

## 3. Electrochemical Sensing

Electrochemical sensing is the most widely used sensing method for all sorts of biosensors, MIP biosensors are without exception, and because of the sensitivity of MIP and the sensibility of the electrochemical sensing method, they have attracted considerable attention to the efforts to combine MIPs with electrochemical sensing to build electrochemical-MIP-biosensors [65,66]. In a standard structure, a MIP nanofilm is polymerized through electrochemical or chemical methods and immobilized on the electrode to fabricate a sandwich-like assembly. The process of producing electrochemical-MIP biosensors is shown in Figure 6. Electrochemical sensing is directly based on the electrochemical signals produced in the sensing process of redox reaction between template molecules and modifiers on the working electrode or the electrode itself, namely the electron transportation of redox reaction. Detailed information about electrochemical sensing with MIP biosensors is summarized in Table 2.

### 3.1. Electrochemical Analytical Method

Voltammetry and impedance are two major ways to monitor the whole fabrication process and reflect sensing behaviors of electrochemical-MIP-biosensors. Voltammetry involves recording the current generated with the varied potential, specifically, it includes Cyclic Voltammetry (CV), Differential Pulse Voltammetry (DPV), and Square Wave Voltammetry (SWV). CV is the most selective method which involves linear potential of triangular waveform, achieved by cyclic sweep scan. Because with the increasing electron transfer, comes higher redox peaks, with the help of detecting the electron transfer between the redox probes, for example [Fe(CN)_6_]^3−/4−^ ions, and the working electrode, one can use CV to monitor the whole fabrication process of the MIP biosensor. Pre-polymerized decoration to the electrode is accompanied with the rising of redox peaks, since grafting modifiers like metal nanoparticles or carbon materials will enhance the conductivity and facilitate electron transportation on electrode interface. On the contrary, the following step by polymerizing MIP nanofilms onto the electrode will suppress the electron transfer and lead to the decrease of redox peaks. But after the extraction of template molecules, the redox peak will rise again because it allows redox probe to get through and participate in the redox reaction again. When sensing and rebinding with analytes, the redox peak will decrease again for the same reason. DPV and SWV are advanced Voltammetry using different applied potential function, which offer higher sensitivity. As for DPV, the potential function is obtained by superimposing a series of regular voltage pulse to a linear sweep voltammetry or a staircase voltammetry, while SWV is combined with a symmetric square wave and staircase potential, the plotted current is measured by subtracting the reverse current waveform from the forward current waveform, since the plotted current is higher than either of the forward or the reverse current, SWV offers even higher sensitivity than DPV. Furthermore, Electrochemical impedance spectroscopic (EIS) is the most widely used impedance method in this field, the EIS includes a semicircle portion at high frequency and a linear portion at low frequency, which corresponds to the electron-transfer restriction and diffusion process.

### 3.2. Electrochemical Sensing Mechanism

To evaluate the performance of electrochemical-MIP-biosensors, one can use the same analytical methods as discussed above. However, to illustrate the sensing mechanism, basically we need to classify it into three separately mechanisms. In the first case, the redox current is generated by the redox reaction between the analytes and the electrode after the rebinding of template molecules to the biosensor. Under this mechanism, with the increasing analyte concentration, more rebinding will cause a stronger redox reaction and lead to an increase of the peak. In another case, also named “gate-controlled” mechanism or the “gate effect”, the redox current could be produced by the redox reaction between the redox probes and the electrodes, redox probes in the solution will pass through the MIP nanofilms to react with the electrode when template molecules are absent or extracted. As a result, it is obviously that the increasing analytes concentration will lead to the decrease of current peak, since template molecules impede the electron transportation between redox probe and the electrode. Furthermore, the competitive mechanism, is to employ a structurally similar analyte derivative to compete with the analyte, the electroactive competitors always bind with the MIP film first, which induce an initial current peak, after adding the analyte, current peak is changed because of the competitive mechanism between the analyte and the competitor.

### 3.3. Electrochemical-MIP-Biosensors

In the last decade, most works have been dedicated to enhancing the performance of electrochemical-MIP-biosensors, namely the sensitivity of the biosensor. Various sorts of new designs have emerged, but to generalize, the overall ideology is to improve the biosensor’s conductivity, specifically focus on two facets:1)Conductivity of the MIP nanofilm, which refers to the selection and polymerization of functional monomers.2)Conductivity of the transducer, which refers to the selection of the electrode material as well as modifiers decorated on the electrode.

Electrochemical polymerization is the traditional and classical method for synthesizing electrochemical-MIP-biosensor, which is a straightforward means of obtaining MIP films with a certain thickness by controlling the number of cycles or the current that is applied to the electrode as discussed above. In recent years, some photo-induced polymerization methods have been combined with electrochemical polymerization for synthesizing electrochemical-MIP-biosensor, such as UV-polymerization and surface-initiated photopolymerization, which are simple, fast and versatile; they provide MIP nanofilms with well characterized chemical structures. Table 2 shows detailed information of the synthesis approach and imprinting techniques to produce MIP biosensors for electrochemical sensing.

#### 3.3.1. Electroactive MIP Nanofilms

Classic organic polymeric materials, especially functional monomers, for instance acrylic acid, vinyl acetate, and pyrrole, offer great advantages of easy manufacture, decent reproducibility and ease of controlling the thickness of MIP nanofilms through electropolymerization. Nevertheless, it is obviously that these classic functional monomers are electric insulators, which is detrimental to the electron transport, so needless to say other organic polymeric materials are involved in the process such as the cross linker and the initiator. Under such circumstances, exploiting new displacements has become a research hotspot in recent years, and several studies have shown that some electroactive functional monomers can be involved and manage to participate in polymerizing MIP nanofilms, which means MIP nanofilms can be employed as a contributor rather than a blocker of biosensor conductivity. Aryl compounds and their derivatives have drawn great attention, and several studies have managed to enhance the MIP conductivity with the use of methacrylic acid [72,93,96], and an aryl diazonium salt, which offers stable aryl bonds. In another study, *N*-acryloyl-4-aminobenzamide [85] was applied as the functional monomer polymerized with graphene quantum dots; Furthermore, other studies indicated that the involvement of Prussian Blue [70] serves the same purpose as electron transfer facilitator. CV has demonstrated the contribution of these functional monomers to the conductivity of the biosensor by showing a clearly increasing current peak after MIP polymerization. Lakshmi [69] designed a new monomer, *N*-phenylethylenediamine methacrylamide (NPEDMA), which has the intrinsic electron transport nature and simultaneously possesses both aniline group and methacrylamide functions. After electro-photo polymerization with NPEDMA, the MIP nanofilm serves as an electroactive “molecule wire” and thus extremely enhances the sensitivity of the electrochemical-MIP biosensor.

#### 3.3.2. Selection of Electrodes

Selection of proper electrodes is an indispensable step for any conductivity enhancement since the intrinsic properties of electrode materials play a decisive role in the overall transducer conductivity.

Noble metals are eligible for electrode materials, among which Au [67,68,69,72] and Pt [70] are the most widely used. Noble metals possess excellent electron transfer properties and a large anodic potential range. Thiol groups can be easily grafted to the gold electrode surface, which facilitates further decorations of the electrode. However, metal oxide films always form at the surface on electrodes which induces a strong background current and interferes with electrochemical analysis. Other metals such as copper wire [89] are also selected to be electrode. Song [71] synthesized nanoporous Au-Ag alloy microrods as the working electrode, which have extremely large surface areas for enhancing electron transfer and show extremely low LOD of 2.7 × 10^−14^ M for detecting metronidazole.

Carbon-based electrodes are more widely used due to their wide potential window, good chemical inertness, rich surface properties, low background current and low price. Glassy carbon electrodes (GCEs) are the most frequently applied carbon-based electrodes, since besides the superior properties listed above, GCEs possess high hardness and good electrical conductivity. Other carbon-based electrodes such as pencil carbon electrode [91,92], ceramic carbon electrode [79], and screen-printed carbon electrode [85] have also been involved in fabricating electrochemical-MIP biosensors in recent years. Nevertheless, the biggest drawback of carbon-based electrodes is their electron transfer rate compared with metal electrodes. As a result, decorations and modifications of electrodes have become a necessity for new era biosensors.

#### 3.3.3. Decorations and Modifications to Electrodes

Metal nanoparticles are involved in electrode decoration and modification due to their good biocompatibility, and large specific surface area which leads to high conductivity. Especially for Au naonoparticles (AuNPs), plenty of works [73,74,75] employed AuNPs also because it is easily grafted to the surface with the control of Au-S bonds through silanes and thiols. Some modified Au NPs were also reported to be involved such as cubic AuNPs [74] as shown in Figure 7A, and Au microflowers [75] formed by using bionic polydopamine film to wrap AuNPs as shown in Figure 7B, which could not only enhance the conductivity but also help firmly immobilize MIP films. Other metal NPs including Pt [82] and Ni [76,77,86] NPs are also investigated for decorations and modifications to electrodes. To extremely enhance the conductivity, hollow nickel nanospheres [76] (as shown in Figure 7C) and 3D nanoporous nickel skeleton [77] were fabricated and attached to GCE and gold electrodes surfaces, respectively, as a result of their high surface area where electron transfer is further facilitated.

A number of sp^2^ hybridized carbonaceous nanomaterials have drawn more and more attention for functional decorations and modifications to electrodes [97], including 0D graphene quantum dots (GQDs) [26,34,85,86] and fullerene [98], 1D carbon nanotubes (CNTs) [79,80,81,82,88,90,99], 2D graphene [75,78,88] and graphene oxide sheets [74,87,89]. Semiconductor quantum dots are mainly used for fabricating bioprobes which function on optical signals, but some recent studies also employed QDs, such as CdS QDs [84], for electrochemical-MIP biosensors, where they mainly play the role of “amplifier” for electron transfer. Graphene quantum dots (GQDs) are a new generation of nontoxic QDs [100] which not only serve as electron transportation pathways, but also possess carboxylic moieties at the edge which help water dispersibility and facilitate the reactions of monomers and other modifier compounds with each other.

Graphene and graphene oxide nanosheets are used because of their large surface area with low dimensions, fast electron transportation and good absorption capacities, especially for graphene oxide, the sharp edge of which was found to be capable of destroying bacteria, and has abundant functional groups and moieties that also ensure graphene oxide is an eligible anchor for other modifiers. Carbon nanotubes are also being widely utilized for the same purpose, including single-walled carbon nanotubes as well as multi-walled carbon nanotubes. Furthermore, other novel carbonaceous derivatives are introduced to the area, for example, highly ordered mesoporous carbon [83] with large pore volume offering a great surface area; Graphitic carbon nitride (g-C_3_N_4_) [82] with C-N bonds which has atomic-scale thickness and facilitates electron transfer.

Neither metal nanoparticles nor carbonaceous nanomaterials are expected to be used exclusively, but rather combined with other sorts of modifiers to fabricate compounds which will always lead to good results beyond expectations and even endow the sensing process with new functions. To illustrate, graphene sheets help immobilize and stabilize metal NPs, while NPs enhance the system conductivity and prevent graphene sheets from face-to-face aggregation. Rao [86] mixed hollow Ni nanoparticles with graphene quantum dots, Yola [74] let cubic AuNPs work with graphene oxide while Yang [75] decorated Au microflowers onto graphene. Not limited to the function of improving electron transportation rates, the metal NPs-carbonaceous nanomaterials model has already been justified of great significance for photothermal therapy [87,101], which is another motivation for compounding metals and carbonaceous nanomaterials. For example, CNTs have been proved to be capable of killing bacteria through a photothermal process [102], Wang [80] and Ji [81] both fabricated AuNPs grafted to MWCNTs for detecting olaquindox and cholesterol, respectively, through which Ji achieved impressive low LOD of 3.3 × 10 ^−14^ mol·L^−1^. G-C_3_N_4_ is also revealed to be helpful to photothermal therapy [103], but the pristine g-C_3_N_4_ still lacks electron transfer ability, so to advance the performance, Yola [82] fabricated g-C_3_N_4_ nanotubes and managed to embed Pt NPs on g-C_3_N_4_ nanotubes and electropolymerized them on a GCE electrode for detecting atrazine at a very low LOD of 1.5 × 10^−13^ M. In recent years, studies have transferred their focus on replacing classic metal NPs modifiers to bimetal, alloy, metal oxide systemd, or the combinations of one with each other. Specific properties will be enhanced for bimetallic NPs due to the synergistic effects between distinct metals. While both Ag and ZnO are demonstrated to individually possess high performance in photothermal therapy, Roy [87] combined Ag-ZnO bimetallic NPs (BMNPs) with graphene oxide to synthesize a Ag–ZnO BMNPs@GO nanocomposite as shown in Figure 8A, and after modification of a GCE electrode, this novel electrochemical-MIP biosensor functioned as three-in-one kit for bacteria detection, removal and killing, where the detection and removal are dependent on the MIP film while the killing is achieved by photothermal therapy. Similar novel systems also include PtAu/CNTs/graphene/GCE [88] as shown in Figure 8B, Fe@AuNPs/MWCNs/GCE [90] as shown in Figure 8C, and MnO_2_/CuO/GO/copper wire [89] as shown in Figure 8D.

Another distinctive compound system is obtained by the involvement of magnetic NPs, especially the most commonly used Fe_3_O_4_ NPs, due to their intrinsic magnetic properties with low cost [104]. There are two general advantages of the magnetic system over other compound systems:(1)It helps with easy control and aggregates analytes which facilitates subsequent collection and further study of analytes.(2)The synthesis procedure is rather simple since Fe_3_O_4_ NPs can be controllably added or removed by an external magnetic field.

So far, there has been several compound structures such as Fe_3_O_4_/MWCNTs/carbon electrode [93,105], Fe_3_O_4_@Au/GCE [94], Fe_3_O_4_/rGO/GCE [95]. Bimetallic magnetic particles are also successfully employed, for instance, the structure of FeAg/RGO/PGE [91] as shown in Figure 9A, or FeCu magnetic NPs/PGE [92] as shown in Figure 9B.

## 4. Gravimetric Sensing

Gravimetric sensing is one of the most mature and traditional working approached for MIP biosensors, which has been demonstrated and improved by numerous studies through the past decades. By detecting the mass change, MIP biosensors transfer firsthand mass signals to analyzable electronic signals, namely through piezoelectric effect using QCM (quartz crystal microbalance). The detailed information of gravimetric sensing with MIP biosensors is summarized in Table 3.

### 4.1. Piezoelectric Sensing Mechanism

A quartz crystal microbalance (QCM) is a supersensitive gravimetric detector based on the piezoelectric effect, which can be used to precisely detect minute mass changes through the variation of the QCM frequency [106,107,108,109,110,111]. Bearing preponderant sensibility but limited by its own selectivity, QCM can be readily combined with molecular imprinting methods, to date, always a molecularly imprinted film, to build a QCM-biosensor. Generally, a QCM-biosensor is fabricated as a micro-sandwich structure, with a Au or Ag electrode coated on both sides of a quartz crystal oscillator as shown in Figure 10. First, the gold electrode should be treated with thiols, which can form an ultrathin thiol film to ensure strong links between the gold electrode and the imprinting polymeric film through –SH bonds. Basically, the imprinting method can be divided into surface imprinting with soft lithography (stamping) and micro-contact imprinting with whole cells. The former presses the stamp directly into the achieved polymeric film to build cavities by using the pressure of template molecules. This method applies to a wide range of both microbiomolecules of macrobiomolecules. The latter one requires immobilization of template molecules on the substrate in advance of polymerization, hence, cavities form within the polymerization process when polymers avoid template sites. This method applies mainly to macrobiomolecules. Table 3 shows detailed information of the imprinting techniques used to produce MIP biosensors for gravimetric sensing.

Based on the piezoelectric effect of quartz crystal oscillators, when binding or rebinding with template molecules, the surface mass change results in a piezoelectric effect in the QCM, leading to a crystal oscillation frequency shift, or to be specific, a decrease of the frequency, following the Sauerbrey equation:$$\Delta f=-2.6\times 10^{6}f_{0}^{2}\Delta m/A$$
where f_0_ is the resonant frequency (Hz), Δm is the mass change, *A* is the piezoelectrically active crystal area, and Δf is the frequency change of QCM.

### 4.2. Gravimetric Sensing Application

QCM has a relatively rich history compared with other materials fabricated with MIPs, as the usage of QCM-biosensors can be certainly dated back to the 1980s. To date, the applications of QCM-MIP-biosensors have been widely used for different types of biological substances from micro-biological molecules like glucose [112], to macro-biological molecules, for instance, DNA [113], proteins [114,115,116], bacteria [117,118] and virus [119], furthermore it has been reported that whole cells can be also imprinted for QCM-MIP-biosensors [120,121].

Plenty of studies using QCM-MIP-biosensor sensing for micro-biological molecules have been reported since the end of the last century. Sensing for small molecules, such as monosaccharides, hormones, or glucose, is relatively simple for QCM sensors considering the easy imprinting and coating process for small molecules. Using QCM sensors, Ersöz [112] managed to detect glucose and achieve continuous glucose monitoring; the group controlled the LOD under 0.07 mM, which represents better sensing ability than traditional glucose binding proteins.

Unlike the ease of imprinting small biological molecules, it is much more difficult to imprint macro-biomolecules like DNA, proteins, bacteria and virus, and especially whole cells. As mentioned above, MIP selectivity is attributed to both physical factors and chemical moieties, so imprinting larger analytes is more contingent on both the physical shape of MIP cavities and chemical recognition of binding ligands. However, in recent years, more and more studies show that the obstacles have been gradually overcome, as researchers have managed to build QCM-MIP-biosensors by imprinting large biomolecules on whole cells. On the other hand, QCM itself is more suitable for sensing large biomolecules instead of small ones, because when sensing micro-biomolecules, mass changes are always incapable of generate sufficient electrical signals, which will result in inaccuracies.

Chunta [116] fabricated QCM-MIP-biosensors for lipoprotein, and high selectivity was demonstrated through a competitive assay, with the LOD of 4 mg/dL. Ma [114] built a QCM biosensor for the epitope of human serum albumin; the LOD can be controlled under 0.026 μg mL^−1^, which is comparable or even better than some electrochemical biosensors. Lee [115] managed to detect three different proteins simultaneously by a QCM-MIP-biosensor with the LOD of 0.1 mg/mL. The sensor can be used in real mixed protein samples and achieves comparable effectiveness to that of a commercial ARCHITECT ci 8200 chemical analysis system. Poller [118] succeeded in building a QCM sensor for *E. coli*, while Yilmaz [117] compared the QCM *E. coli* sensor to the SPR one as shown in Figure 11A, demonstrating a smaller LOD and LOQ of 3.72 × 10^5^ CFU/mL and 1.24 × 106 CFU/mL respectively, and a shorter response time of 56 s. Studies for detecting other analytes including viruses and whole cells, for example yeasts [120,121], substantiate the broad applicability of QCM-MIP-biosensors.

Traditional QCM biosensors have been maturely combined with MIP for sensing different analytes. However, the structures of biosensors still need continuous improvements to further eliminate errors and enhance their sensitivity. Traditional QCM biosensors require two separate quartz crystals in order to measure MIP and NIP, respectively, resulting in larger sensor size, and an increased price tag; moreover, crystal-induced errors are magnified by the non-identical crystals of MIP and NIP due to the imperfect fabrication process, which leads to further measurement errors. Croux [122] managed to solve the problem by combining four separate electrodes on a single piece of quartz to build a 4-channel QCM-MIP biosensor as shown in Figure 11B. Based on finite element analysis modeling and the introduction of mesa-like structures on the crystal surface, crosstalk of frequency coupling between four channels are successfully avoided, and the biosensor was used for detecting L–nicotine with a LOD of around 15 mM.

### 4.3. Gravimetric Sensing with Electrochemical Methods

Gravimetric techniques based on piezoelectric effect are still incapable of supplying sufficient sensing signals, and some studies have revealed that the LOD using a gravimetric technique alone is much higher than that of using electrochemical sensing under the same conditions. Consequently, in recent years, gravimetric sensing has begun to be used to assist with electrochemical analysis methods such as CV [98,123,124], SWV, DPV [125], and EIS [126,127,128] to further enhance the sensitivity of MIP biosensors. Besides, more modifiers will be available if the gravimetric technique is combined with electrochemical methods. The structure of the biosensor is virtually identical to that of electrochemical-MIP biosensors discussed above, except that the substrate here is a QCM. Kong [127] grafted AuNPs to the QCM while Fang [128] constructed AuNPs on mesoporous carbon CMK-3 and used the composite to modify the electrode. Sharma [98] described an innovative representative structure by employing three distinct fullerene derivatives as functional monomers to polymerize with Pt NPs for the detection of adenosine-50-triphosphate as shown in Figure 11C. While a few works report using fullerene for fabricating MIP bioprobes [33,129], this work was the first attempt to involve fullerene in producing MIP biosensors and has been the only one up to now. The LOD in this work was 0.3 and 0.03 nM, based on piezoelectric microgravimetry and capacity impedance respectively, which indicates that there are still many opportunities to enhance gravimetric sensing.

## 5. Magnetic Sensing

Magnetic sensing is an emerging approach that combines magnetic analysis methods with molecular imprinting technology. By means of magnetometry, magnetic-MIP bioprobes or biosensors enable one to reflect specific magnetic properties including saturation magnetization, paramagnetism, remanence, etc. So far, the applications of magnetic sensing have mainly focused on magnetic-MIP bioprobes.

### 5.1. Magnetic-MIP Bioprobes

In a classic structure, magnetic-MIP bioprobes are fabricated as core-shell NPs, with magnetic NPs, especially Fe_3_O_4_ NPs, as the core substrate, and a MIP film as the shell. Figure 12 shows the production process of a magnetic-MIP bioprobe. Detailed information about magnetic sensing with MIP bioprobes is summarized in Table 4.

Magnetic-MIP bioprobes have obvious advantages over other devices in several aspects:1)The magnetic MIP composite allows external-magnetic-field isolation and enrichment of analytes, which extremely enhances the bioprobe’s capability to not only detect analytes, but also aggregate analytes.2)Magnetic NPs preserve the advantages of NPs such as high surface area which is favorable for the selectivity.3)Magnetic-MIP bioprobes exhibit significant durability, permitting good reusability for more times.

Magnetometry shows that the saturation magnetization will decrease after Fe_3_O_4_ are covered with a MIP film, which is ascribed to the NP surface effects such as disorder in magnetically inactive films in oposition to the total magnetic field [104], however, the saturation magnetization is sufficient to enhance the adsorption of analytes. Gao [130] fabricated a Fe_3_O_4_@MIP magnetic bioprobe for bovine hemoglobin as shown in Figure 13B, where magnetometry showed that the saturation magnetization reached 25.47 emu/g, which helped the amount of absorbed analytes reach 110.5 ± 0.83 mg/g. The bioprobe was proved to capable of recycling at least six times. Kan [131] employed the same structure for the same analytes and demonstrated that the adsorption of the magnetic-MIP bioprobe was 4.6 times higher than that of a NIP one. Zhang [132] synthesized a bioprobe by using a microwave heating method, which attained an atrazine adsorption amount of 144.0 ± 2.2 mL/g. As shown in Figure 13C,D, the bioprobe was still activate after working 100 times. Still using Fe_3_O_4_ NPs, some works have even combined magnetic sensing and optical sensing together. Magnetic-MIP bioprobes were fabricated by Li [133] based on the FRET mechanism between MIP films and template molecules, where the Fe_3_O_4_ NPs was not only serve as magnetic substrate to enrich the adsorption, but also favor enhanced electron transfer for the FRET process. Zhou [134] devised a Ru@SiO_2_@Au magnetic MIP composite as a bioprobe for hemoglobin.

As shown in Figure 13A, this magnetic-optical-MIP bioprobe displayed an adsorption amount of 264.60 ± 0.04 mg/L for a real sample, and even exhibited optical sensing of increasing luminescence intensity with the increasing analyte concentration.

### 5.2. Magnetic-MIP Biosensors

Magnetic nanomaterials are widely used in MIP biosensors, however, most works barely involve magnetic NPs as modifiers and decorations to the electrode of electrochemical-MIP biosensors as we have discussed above, for the purpose of enhancing conductivity of the transducer. Indeed, works on MIP biosensors functioning by magnetometry are extremely rare. For sensing pyridoxine, Patra [92] fabricated an electrochemical-MIP biosensor with Fe/Cu bimetallic magnetic NPs as modifiers grafted onto the electrode, and magnetometry was applied to ensure the change of magnetic properties in the polymerization process. The result showed a fairly low LOD of 0.040 μg/L by this method. This work is one of the few studies we have found about the involvement of magnetic sensing in MIP biosensors, indicating that the magnetic sensing still has a long way to go to be truly combined with MIP biosensors.

## 6. Critiques and Outlook

Over the last few decades, molecular imprinting technology has attracted great attention, and the rapid development of relevant biological fields has tremendously motivated a continuous involvement of MIP in bioprobes and biosensors. In this work, we have systematically reviewed recent MIP bioprobes and biosensors applied to biomolecular detection and drug tracking from four dimensions, namely, the optical sensing, the electrochemical sensing, the gravimetric sensing and the magnetic sensing. The working mechanisms of each sensing method were thoroughly analyzed. Optical sensing is based on a series of mechanisms including fluorescence quenching and enhancing mechanisms as well as PET and FRET. Electrochemical sensing is based on the “gate-control” mechanism or a counterpart. Gravimetric sensing is based on the piezoelectric effect, while magnetic sensing is based on analyzing magnetic properties. One of the reasons studies focus on MIP rather than traditional approaches such as antigen-antibody binding is that the MIP bioprobes and biosensors are easily decorated for further modifications to produce unique structures. We have reviewed the types of modifiers that have emerged in recent years, including the original functional monomers, metal nano-NPs, and carbonaceous nanomaterials. We can firmly conclude that after either of the modifications, or with new-designed structures, a number of changes will be observed in the properties of the MIP bioprobes and biosensors, including enhancements of existing properties or complementary new properties. Indeed, more works have attempted to mix distinct modifications and they did achieve higher performance. Furthermore, the desire to unite different sensing methods in a single study is getting stronger. This review discusses examples of the combination of optical sensing with magnetic sensing, electrochemical sensing with gravimetric sensing, and electrochemical sensing with magnetic sensing.

Challenges and pending concerns still exist with the development of MIT in both sample studies and market expansion. For sample studies in the laboratory, first, the issue of simultaneously attaining high selectivity and high sensibility must be addressed. Combining MIT with bioprobes and biosensors is the original idea of attaining both high selectivity and sensibility, after that, several approaches had been applied to go further, including the decoration and modification to transducers and some new structures. However, we need more than that. New synthesis approaches are urgently needed for thicker and mesoporous shells of MIP bioprobes and thinner films for biosensors; The signal change for optical sensing to trace template molecules should be large enough to be observable even by the naked eye. Second, as mentioned above, sensing for macromolecules such as proteins, DNA, cells, bacteria, viruses will continue to be a serious issue, especially for optical sensing by SPR, and gravimetric sensing by QCM. Different imprinting strategies must be continuously developed for synthesizing MIP films. Computational or combinatorial tools are also expectedly adopted for preparing “smart” MIP films. Third, detection of small molecules, especially gaseous molecules, is rarely attempted, because of the too small size of molecules, the gaseous state at room temperature and uncontrollable operation parameters. As the applications become broader, gaseous molecule detection and sensing will become the next challenge to be solved. Unlike the widespread application in real samples, large scale production of MIPs for the market is rarely reported, and the huge difference between lab-scale production and industry-scale production of MIP is mainly attributed to the different synthesis conditions, production amounts, total costs, and environmental factors. The issue can be alleviated with a few of the proposed methods by using existing mature synthesis methods, though sacrificing selectivity, or by developing intelligent preparation methods which enable industry-scales production. The commercial exploitation, though still in its infant phase, possesses huge potential in the future.

Despite the limitations, the prospective outlook of MIT should never be ignored. As the introduction of molecular imprinting technology to bioprobes and biosensors gradually becomes mature, researchers are not satisfied with the sole function of biosensing. Instead, more and more studies are trying to develop prototypes into multi-functional kits which are practicable for much broader applications. For example, as we have reviewed above, MIP biosensors have been successfully combined with photothermal therapy to fabricate a three-in-one kit providing sensing, enrichment and killing of bacteria. Indeed, the application of MIP bioprobes and biosensors has gone far beyond simple biomolecule detection and drug tracking, and they manage to be even applied to food engineering and environmental engineering, for food assays and environmental monitoring, respectively. Moreover, there is a clear trend of updating 2D MIP biosensors to 3D bioMEMs. Cutting-edge bioMEMs originated from MIP biosensors, such as MIP electronic tongues [136] and MIP handheld analyzers [137], are continuously appearing in recent years. Electronic tongues are usually an array of low-affinity recognition units. By merging with MIT, the MIP electronic tongues permit simultaneous sensing and discrimination of several analytes in complex solutions with more reliable analytical signals. High-performance liquid chromatography and ion mobility spectrometry are two traditional methods employed for clinically detecting propofol concentrations in blood in hospitals. These conventional instruments are bulky, expensive, and difficult to access, whereas, the MIP handheld analyzer is an ideal alternative with the advantages of compact size, high selectivity, low cost, rapid response, and single-step detection. Not limited to the lab, MIT has been successfully and closely applied in the market, as more and more universities and institutes cooperate with commercial companies and startups to convert them into products. Columbia BioSystems, Inc cooperated with the University of Maryland [138] to develop a dialysis system consisting of custom synthesized MIPs packaged into plastic discs (cartridges) that are compatible with commercially available hemodialysis devices, which enables hospitals, clinics, and other healthcare organizations to turn existing dialysis systems into virus removal systems capable of lowering the viral load in patients with HIV, hepatitis B, hepatitis C, or other blood-borne viruses. New Jersey–based startup Semorex Inc., founded by the MIP pioneer Wulff [139], is collaborating with a pharmaceutical company to screen a group of the latter’s potential drug leads against an MIP model of a drug target. The Semorex team is even focusing on the development of MIPs for detecting chemical weapons, and they have shown that MIPs can distinguish analogs of chemical warfare agents from insecticides, a feat that available technology cannot readily accomplish. After going through all these references and facts, we can safely conclude that there is a sufficient development niche for MIP bioprobes and biosensors both in the laboratory and the market. Although more challenges must be addressed, we are looking forward to their rising positions in wider fields as well as more profound integration with other technologies to explore the maximum potential in the future.

## Figures and Tables

**Figure 1 sensors-19-00177-f001:**
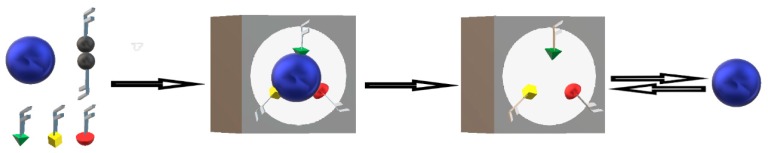
The general process of molecular imprinting technology including the copolymerization of functional monomers and the removal of template molecules.

**Figure 2 sensors-19-00177-f002:**
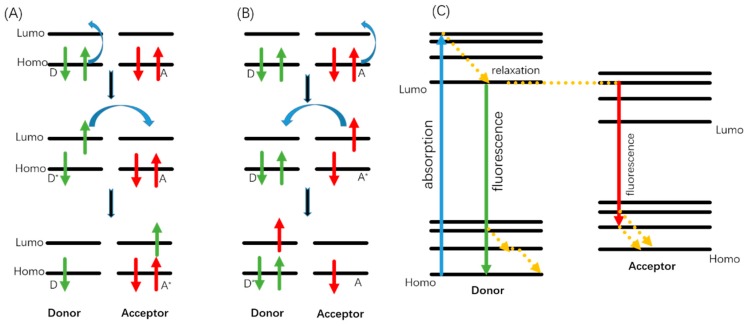
Fluorescence quenching mechanisms. (**A**) Photo-induced reduction mechanism. (**B**) Photo-induced oxidation mechanism. (**C**) Fluorescence resonance energy transfer (FRET) mechanism.

**Figure 3 sensors-19-00177-f003:**
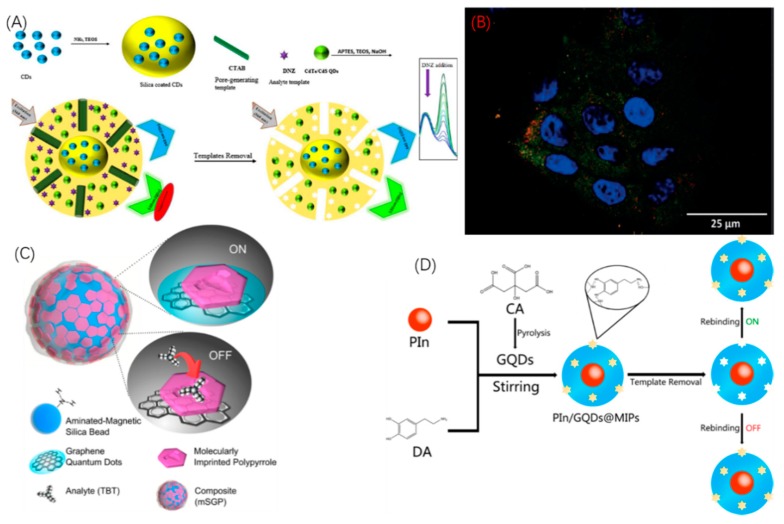
Representative architectures of QDs-MIP bioprobes by fluorescence quenching mechanism. (**A**) Schematic illustration of dual-QDs-MIP bioprobes for DNZ with CDs and CdTe/CdS QDs work simultaneously. (**B**) Confocal microscope image of dual-QDs-MIP bioprobes with InP/ZnS QDs of different two colors staining human keratinocytes simultaneously. GlcA-bioprobe (green), NANA-bioprobe (red), blue areas are nucleus stained by DAPI. (**C**) Schematic illustration of GQDs-MIP magnetic bioprobes for dopamine with GQDs embedded on a magnetic silica bead. (**D**) Synthesis process of dual-QDs-MIP bioprobes for dopamine with GQDs as the core and Pln QDs dispersed in the shell. (**A** is reproduced with permission from [19], **B** is reproduced with permission from [20], **C** is reproduced with permission from [23], **D** is reproduced with permission from [7]).

**Figure 4 sensors-19-00177-f004:**
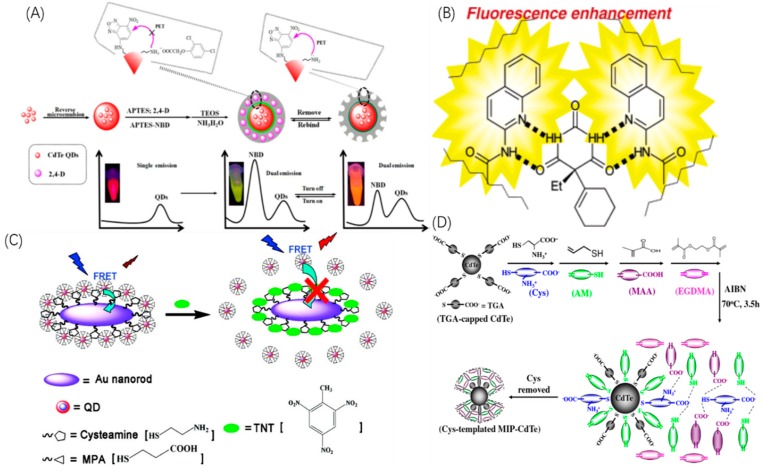
Representative architectures of QDs-MIP bioprobes by fluorescence enhancing mechanism. (**A**) Synthesis process of QDs-MIP bioprobes for 2,4-dichlorophenoxyacetic acid with nitrobenzoxadiazole (NBD) as the fluoresence functional monomer for enhancing PET process. (**B**) Schematic illustration of QDs-MIP bioprobes for cyclobarbital with the functional monomer nitrobenzoxadiazole (NBD) as signaling receptors for enhancing PET process. (**C**) Schematic representation of the QDs-MIP bioprobe by means of FRET from QDs (energy donor) to Au NPs (energy acceptor). (**D**) Synthesis process of CdTe QDs-MIP bioprobes for cysteine under the surface passivation system. (**A** is reproduced with permission from [24], **B** is reproduced with permission from [29], **C** is reproduced with permission from [30], **D** is reproduced with permission from [33]).

**Figure 5 sensors-19-00177-f005:**
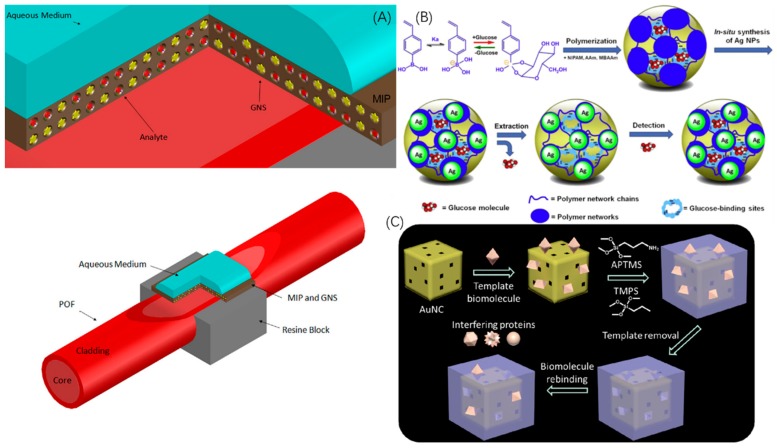
Novel architectures of SPR-MIP bioprobes and biosensors. (**A**) Schematic representation of the SPR-MIP biosensor with tapered optical fibers and decorated with five-branchedgold nanostars in MIP film. (**B**) Schematic illustration of SPR-MIP bioprobes for glucose with Ag NPs surrounded by poly(NIPAM-AAm-VPBA) to form the microgel which control the distance of Ag NPs with each other and ensure them to display SPR property. (**C**) Schematic representation of QDs-MIP gold nanocages, (**A** is reproduced with permission from [47], **B** is reproduced with permission from [49], **C** is reproduced with permission from [50]).

**Figure 6 sensors-19-00177-f006:**
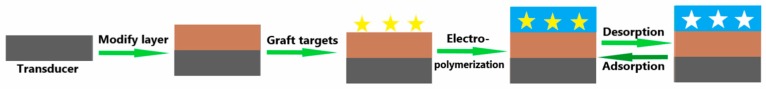
Schematic configuration of the building process of an electrochemical-MIP biosensor.

**Figure 7 sensors-19-00177-f007:**
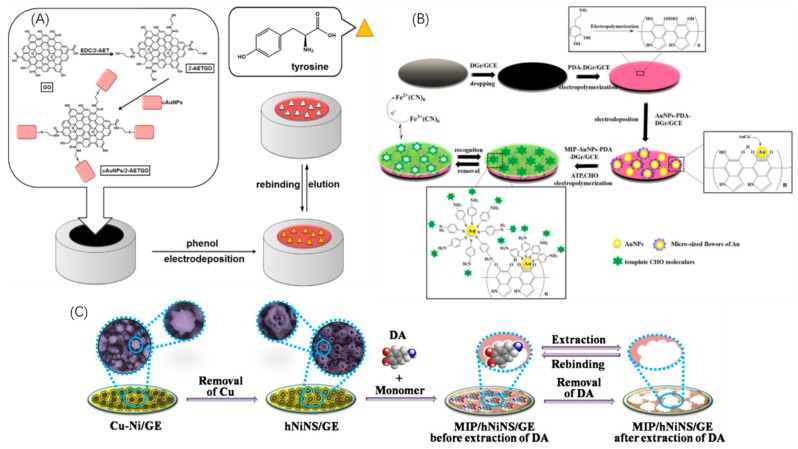
Representative metal NPs in electrochemical-MIP bioprobes and biosensors. (**A**) Synthesis process of decorating cubic Au NPs on GCE. (**B**) Synthesis process of decorating microflowers Au NPs on GCE. (**C**) Synthesis process of cultivating hollow nickel nanospheres on GCE by extracting Cu. (**A** is reproduced with permission from [70], **B** is reproduced with permission from ref. [71], **C** is reproduced with permission from [75]).

**Figure 8 sensors-19-00177-f008:**
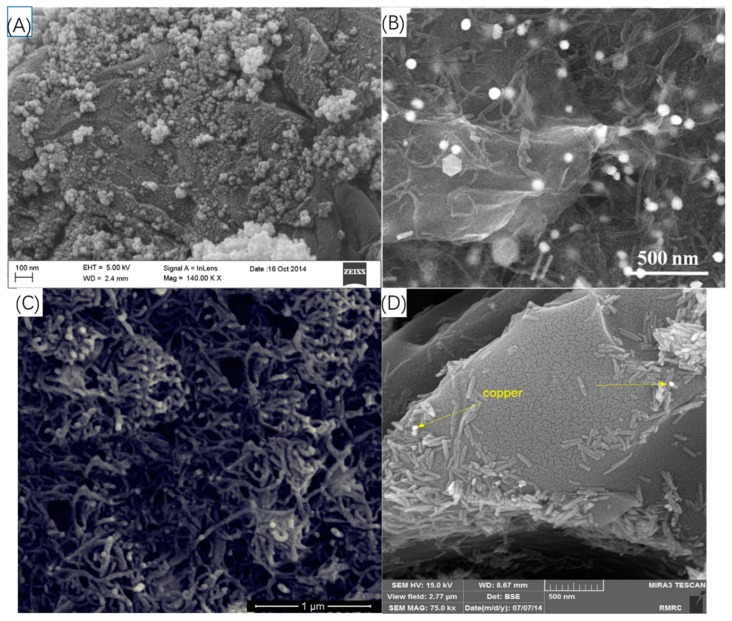
SEM images of bimetallic-carbonaceous composites as decoration on electrode. (**A**) The SEM image of Ag-ZnO bimetallic NPs decorated on graphene oxide, for building the Ag–ZnO BMNPs@GO nanocomposite on GCE. (**B**) The SEM image of PtAu NPs decorated on CNTs and grafted to graphene sheets for building the PtAu/CNTs/graphene/GCE. (**C**) The SEM image of Fe@Au NPs modified on 2-AET functionalized MWCNT for building the Fe@AuNPs/MWCNs/GCE. (**D**) The SEM image of CuO and MnO2 loaded on graphene sheets for building the MnO_2_/CuO/GO/copper wire. (**A** is reproduced with permission from ref [84], **B** is reproduced with permission from [82], **C** is reproduced with permission from [79], **D** is reproduced with permission from [65]).

**Figure 9 sensors-19-00177-f009:**
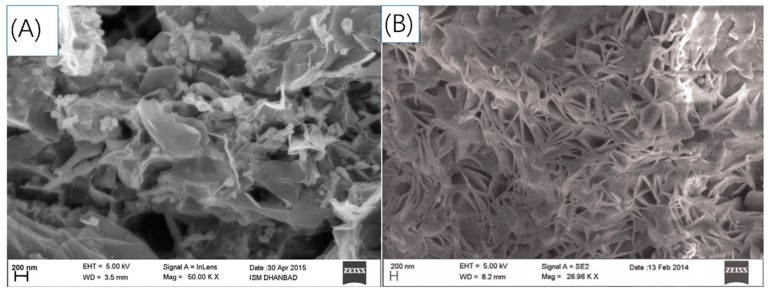
SEM images of magnetic bimetallic NPs decorated with carbonaceous materials for building electrodes. (**A**) The SEM image of decorating magnetic bimetallic FeAg NPs on reduced graphene oxide, for later building the composite on pencil graphite electrode. (**B**) The SEM image of magnetic bimetallic FeCu NPs with MIP modified on the pencil graphite electrode. (**A** is reproduced with permission from [67], **B** is reproduced with permission from [68]).

**Figure 10 sensors-19-00177-f010:**
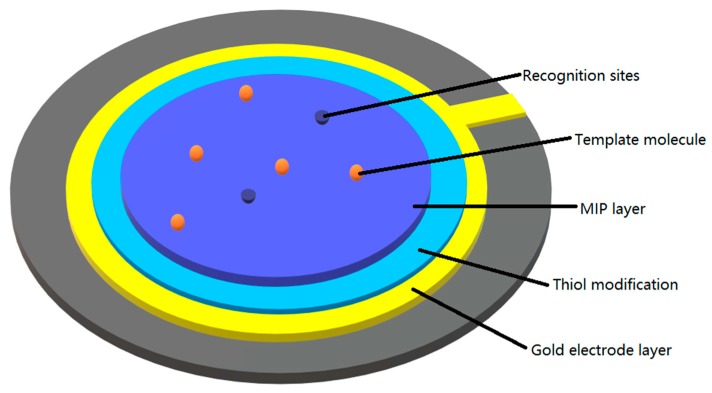
Schematic configuration of the QCM-MIP biosensor.

**Figure 11 sensors-19-00177-f011:**
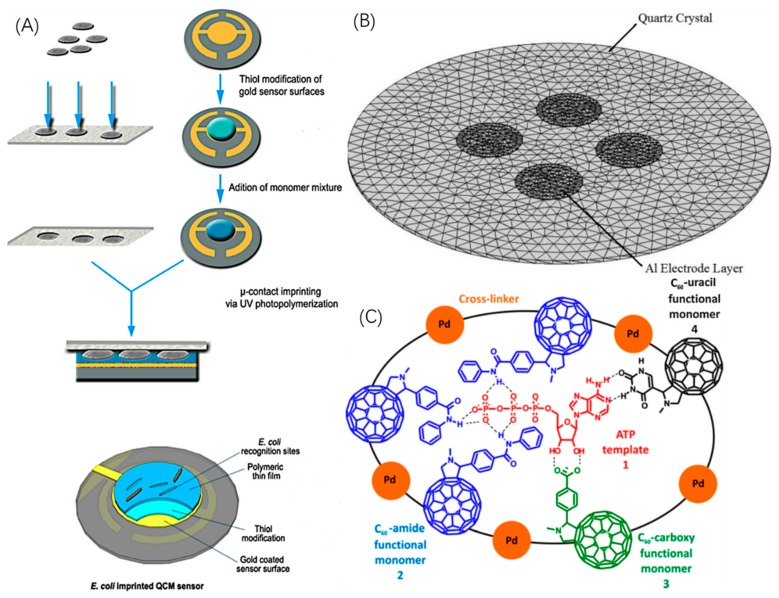
Classic and original structures of MIP-QCM biosensors. (**A**) Synthesis process of the QCM-MIP biosensor by contact imprinting, and representative scheme of the finished *E. coli* imprinted QCM-MIP biosensor. (**B**) Multi-chanel QCM (four channels)‘s finite element model. (**C**) Schematic illustration of the MIP structure decorated on QCM with the template 1 and three different fullurene derivatives 2,3,4 as functional monomers. (**A** is reproduced with permission from [100], **B** is reproduced with permission from [105], **C** is reproduced with permission from [78]).

**Figure 12 sensors-19-00177-f012:**
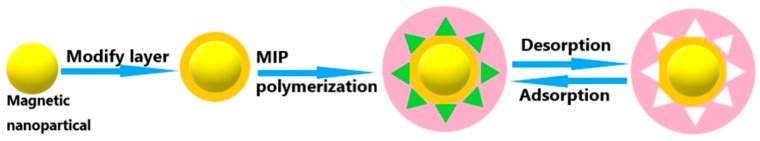
Schematic configuration of the producing process of a magnetic-MIP bioprobe.

**Figure 13 sensors-19-00177-f013:**
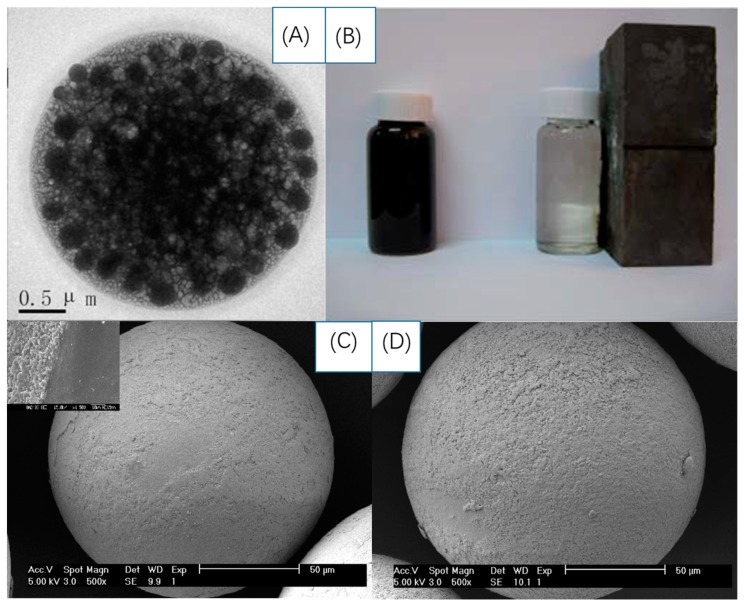
Images of magnetic-MIP bioprobes. (**A**) TEM image of magnetic Fe_3_O_4_-MIP bioprobes. (**B**) Separation process of Fe_3_O_4_-MIP bioprobes with an external magnet. The SEM image of magnetic MIP bioprobes with Fe_3_O_4_ NPs wrapped by PEG (**C**) and after reusing by 100 times (**D**). (**A** is reproduced with permission from ref [117], **B** is reproduced with permission from [113], **C** and **D** are reproduced with permission from [115]).

**Table 1 sensors-19-00177-t001:** Overview of different MIP bioprobes with optical sensing.

QDs	Template	Fabrication Form	Synthesis Approach/Imprinting Techniques	Analytical Method	Size	Emission Wavelength	Linear Range	LOD	Ref
CdTe	Amoxicillin	Traditional form	Sol-gel/Sol-gel copolymerization	Fluorescence microscopy/SEM/TEM/FTIR	2.5 nm	400–700 nm	0.20–50.0 μg L^−1^	0.14 μg L^−1^	[10]
Mn-doped ZnS	Cocaine and metabolites	Traditional form	Ultrasound irradiation/precipitation polymerization	Spectrofluorimetry/fluorescence spectrometry/XRD/FTIR	1.66 nm	400–800 nm	50–400 mg L^−1^		[11]
InP/ZnS	glucuronic acid (GlcA)/N-acetyl-neuraminic acid (NANA)	Traditional form	Qds internal excited photopolymerization	TEM/DLS/epifluorescence microscopy/confocal microscopy	125 ± 17 nm	550 nm, 660 nm			[12]
CdSe SiO_2_	Cyhalothrin	Enveloped by SiO_2_	Modified reverse microemulsion method	Fluorescence spectrometry/SEM/TEM/IR spectroscopy	90 nm	550–700 nm	0.1–1000 μM	3.6 μg L^−1^	[13]
CdTe SiO_2_	Sulfadimidine	Enveloped by SiO_2_	Sol-gel	Fluorescence spectrometry/FTIR/TEM/XRD	130 nm	550–600 nm	10–60 μM	1.9–3.1%	[14]
FeSe SiO_2_	Cyfluthrin in sendiments and fish samples	Enveloped by SiO_2_	Modified reverse micro-emulsion method	Fluorescence spectrometry/SEM/TEM/FTIR	100 nm	435–465 nm	0.010–0.20 mg L^−1^	1.3 µg kg^−1^; 1.0 µg kg^−1^	[15]
CdTe/CdS	Diniconazole (DNZ)	Enveloped by SiO_2_	Sol-gel	fluorescence spectrophotometry/TEM/SEM/FTIR	100 ± 10 nm	530 and 440 nm	20–160 µg L^−1^	6.4 µg L^−1^	[16]
Graphene SiO_2_	Metronidazole	Enveloped by SiO_2_	Sol-gel polymerization	luminescence spectrometer/FTIR/SEM/TEM	100 nm	450 nm	0.2–15 μM	0.15 μM	[17]
CdTe SiO_2_	2,4-Dichloro-phenoxyacetic acid	On SiO_2_ nanosphere	Stöber method/sol-gel polymerization	TEM/FTIR/Spectro-fluorimetry/thermogravimetry	50–80 nm	370–670 nm	0.66–80 μM	2.1 nM	[18]
Graphene SiO_2_	Tributyltin	On SiO_2_ nanosphere	In-situ polymerization	FTIR/XPS/SEM/TEM/Raman images and spectra/AFM	2.37 ± 0.39 nm (GQDs)200 nm to 1.2 µm (mSB)	470 nm (GQDs)460 nm (mSGP)	0.0−10 ppm	12.78 ppb (water)42.56 ppb (seawater)	[19]
CdTe	Tetracycline	Molecularly imprinted glass	Sol-gel polymerization	Spectrofluorimetry/SEM/FTIR	70 μM–2.2 mM	535–560 nm	70 μM–2.2 mM	2.1 μM	[20]
Mn^2+^-doped ZnS	Lysozyme	Molecularly imprinted membrane	Sol-gel polymerization	Spectrofluorimetry/TEM/XRD/FTIR/AFM	3–4 nm (diameter) 15.2 nm (thickness) 1.55 nm (roughness)	350–700 nm	0.1–1.0 μM	10.2 nM	[21]
CdSe/ZnS	Histamine	Molecularly imprinted nanofibers	Organogelation process/in-situ polymerization	Spectrofluorimetry/FTIR/SEM/TEM	50 nm (diameter)	330 nm, 660 nm	100–700 μg L^−1^		[22]
CdTe	S-fenvalerate	On paper-based devices	Wax printing/screen printing	FTIR/SEM/TEM/EIS/EDS/CV	3–5 nm (QDs)/100 nm (MIPs)		10 nM–1 μM	3.5 nM	[23]

**Table 2 sensors-19-00177-t002:** Overview of different MIP biosensors with electrochemical sensing.

Functional Monomers	Template	Method of Polymerization	Electrode	Electrochemical Analytical Method	Linear Range	LOD	Ref
*o*-PD	GSH/GSSG	In-situ electrochemical	Au	CV/EIS	0.04–20 nM(GSH)/0.04–10 nM(GSSG)	1.33 × 10^−2^ nM	[67]
MAH/HEMA	*E. coli*	UV-polymerization/micro-contact imprinting method	Au	CV/capacitance testing	1.0 × 10^−2^–1.0 × 10^−7^ CFU/mL	70 CFU/mL	[68]
NPEDMA	Catechol/dopamine	Electrochemical/photochemical	Au on glass	CV	228 nM–144µM	228 nM	[69]
Prussian blue	Oxytetracycline	Electrochemical	Pt	CV/DPV	0.1–1.0 μM	230 fmol/L	[70]
*o*-PD	Metronidazole	Electrochemical	Nanoporous Au-Ag alloy microrod	CV	8.0 × 10^−5^–1.0 × 10^3^ nM	2.7 × 10^−5^ nM	[71]
Methacrylic acid	Dopamine	Surface-initiated photopolymerization/electrochemical	AuNPs/Au	SWV	0.1–10 nM	0.35 nM	[72]
ATP	CPF	Electrochemical	AuNPs/GCE	CV/EIS	50–100 μM	25 μM	[73]
Phenol	Tyrosine	Electrochemical	cAuNPs/GO/GCE	CV/EIS/DPV	1.0–20.0 nM	0.15 nM	[74]
Aminothiophenol/dopamine	Cholesterol	Electrochemical	Au microflowers/graphene/GCE	CV/EIS/DPV	10^−9^–10^−4^ nM	3.3 × 10^−10^ nM	[75]
Pyrrole/*o*-PD	Dopamine	Electrochemical	hNiNS/GCE	CV/EIS	5 × 10^−5^–5 × 10^−2^ nM	1.7 × 10^−5^ nM	[76]
*o*-PD	Metronidazole	Electrochemical	3D nanoporous Ni/Au	CV/EIS	6 × 10^−5^–1.0 × 10^−6^ nM	2 × 10^−5^ nM	[77]
Pyrrole	Bovine serum albumin	Electrochemical	CS/IL/graphene/GCE	CV/EIS/DPV	1.0 × 10^−7^–1.0 × 10^−1^ mg L^−1^	2 × 10^−11^ mg L^−1^	[78]
Chitin	Cholesterol	Chemical	MWCNT/CCE	CV/LSV	1.0 × 10^−2^–3.0 × 10^−1^ μM	1 nM	[79]
*o*-PD	Olaquindox	Electrochemical	AuNPs/MWCNT/GCE	CV/DPV	10.0–200.0 nM	2.7 nM	[80]
Aminothiophenol	Cholesterol	Electrochemical	AuNPs/MWNTs/GCE	CV/DPV	1.0 × 10^−4^ – 1.0 nM	3.3 × 10 ^−5^ nM	[81]
Phenol	Atrazine	Electrochemical	PtNPs/C_3_N_4_NTs/GCE	CV/EIS/SWV	1.0 × 10^−3^–1.0 × 10^−1^ nM	1.5 × 10^−4^ nM	[82]
*p*-ABA/Prussian blue	Metolcarb	Electrochemical	PB/CMK-3/GCE	CV	5.0 × 10^−4^–1.0 × 10^2^ μM	9.3 × 10^−2^ nM	[83]
Pyrrole	Urea	Electrochemical	CdS QDs/Au	CV/EIS/DPV	5.0 × 10^−3^–7.0 × 10 nM	1.0 × 10^−3^ nM	[84]
*N*-Acryloyl-4-aminobenzamide	Ifosfamide	Chemical	GQDs/screen-printed carbon	CV/DPV	0.25–121.35 μg L ^−1^	0.177 μg L ^−1^	[85]
Pyrrole	Bisphenol S	Electrochemical	hNiNS/GQDs/GCE	CV/DPV	0.1–50 μM	0.03 μM	[86]
APTS	*E. coli*	Chemical	Ag-ZnO/GO/GCE	SWV	10–10^9^ CFU mL^−1^	5.9 CFU mL^−1^	[87]
*o*-PD	Propyl gallate	Electrochemical	PtAu/CNTs/graphene/GCE	CV	7 × 10^−2^–1.0 × 10 μM	2.51 × 10^−2^ μM	[88]
Vinyl acetate	Glucose	Chemical	MnO_2_/CuO/GO/copper wire	CV	0.5–4.4 mM	53 μM	[89]
Pyrrole	Cefixime	Electrochemical	Fe@AuNPs/MWCNs/GCE	CV/EIS/SWV	1.0 × 10^−4^–1.0 × 10^−2^ μM	2.2 × 10^−5^ μM	[90]
Acrylic acid	Pyrazinamide	Chemical	FeAg/RGO/PGE	CV/SWV	1.996 to 740.74 ng L^−1^	0.66 ng L^−1^	[91]
Acrylic acid	Py/PLP	Electrochemical	FeCu magnetic NPs/PGE	CV/EIS/SWV	0.099–196.0 μg L^−1^ (Py)/0.199–157.4 μg L^−1^ (PLP)	0.04 μg L ^−1^ (Py)/0.043 mg L ^−1^ (PLP)	[92]
Methacrylic acid	Kanamycin	Chemical	Fe_3_O_4_/MWCNTs/carbon electrode	CV/EIS/DPV	1.0 × 10^−4^–1.0 μM	2.3 × 10^−5^ μM	[93]
Gsh	Estradiol	Chemical	Fe_3_O_4_@Au/GCE	CV/DPV	0.025–10.0 μM	2.76 nM	[94]
Aniline	Amaranth	Electrochemical	Fe_3_O_4_/rGO/GCE	DPV	0.05–50 μM	50 nM	[95]
Methacrylic acid	Chlorotoluron	Chemical	magnetic NiHCF NPs/GCE	CV	5 × 10^−3^ to 1 × 10^−1^ μM	9.27 × 10^−4^ μM	[96]

**Table 3 sensors-19-00177-t003:** Overview of different MIP biosensors with gravimetric sensing.

Functional Monomers	Template	Imprinting Techniques	Linear Range	LOD	Analytical Method	Advantages	Ref
MAH	α-D-glucose	Drop coating	0.07–8 mM	0.07 mM	QCM	Successful involvement of metal coordination (metal–ligand chelate)	[112]
AA/MAA/VP	Low-density lipoprotein (LDL)	Stamp imprinting/spin coating	4–400 mg dL^−1^	4 mg dL^−1^	QCM/AFM	Fast easy and cost-effective testing	[116]
Zinc acrylate	C- terminus epitope of human serum albumin	Drop coating	0.050 μgmL^−1^–0.500 μgmL^−1^	0.026 μg mL^−1^	QCM/AFM/SEM/UV-vis/HPLC	Easy and cheap/good selectivity and sensibility/low-cost, time-saving	[114]
Ethylene/vinylalcohol	Salivary proteins	Thermally induced phase separation (TIPS)	0.29–0.46 μg mL^−1^	0.1 mg mL^−1^	QCM/AFM/ARCHITECT ci 8200 system	Accurate, feasible and economical	[115]
Urethane	Human rhinovirus (HRV)/foot-and-mouth disease virus (FMDV)	Stamp imprinting/soft lithography/spin-coating			QCM/AFM/Brunauer-Emmett-Teller (BET) analysis	Rapid analysis high cross-sensitivity	[119]
Epon1002F	*E. coli*	Surface imprinting/nanoimprint lithography (NIL)/spin-coating/PDMS stamp	0.4–7.3 × 10^7^ CFU mL^−1^	1.4 × 10^7^ CFU mL^−1^	AFM/QCM	Superior sensitivity and signal intensities, Easy reproducibility and further shortening of imprint fabrication time	[118]
HEMA/EGDMA/MAH/AIBN	*E. coli*	Micro contact imprinting	0.5–3.0 McFarland	3.72 × 10^5^ CFU mL^−1^	QCM/SPR/AFM/ellipsometer measurements	Real-time detection capabilities and total detection of 1 h or less and low organism detection limits/high stability and reusability	[117]
Urethane/vinylpyrrolidone	Artificial yeast/erythrocyte	Master stamp/stamp-less imprinting			AFM/QCM	Feasible for process monitoring/high cross-selectivity	[120]

**Table 4 sensors-19-00177-t004:** Overview of different MIP bioprobes with magnetic sensing.

Functional Monomers	Template	Magnetic Analysis Method	Saturation Magnetization Value	Amount of Absorbed Analytes	Ref
Tetraethoxysilane	Bovine hemoglobin	VSM	25.47 emu g^−1^	110.5 ± 0.83 mg g^−1^	[130]
*γ*-Aminopropyltrimethoxysilane/tetraethyl orthosilicate	Bovine hemoglobin	VSM	50 emu g^−1^	10.52 mg g^−1^	[131]
3-(Triethoxysilyl)propyl isocyanate	Estrone	VSM	44.63 emu g^−1^	183.4 μmol g^−1^	[135]
Nitrobenzoxadiazole	Rhodamine B	VSM	37.8 emu g^−1^	29.64 mg g^−1^	[133]
Methacrylic acid	Atrazine	VSM	0.491 emu g^−1^	144.0 ± 2.2 mL g^−1^	[132]

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
