# Peer review of "Advances in Molecularly Imprinting Technology for Bioanalytical Applications"

_sensors, 2019, doi:10.3390/s19010177_

Round 1

Reviewer 1 Report

General comment:

Concerning the strength of the article, the subject is very interesting and provides a comprehensive review of MIP-based bioprobes and biosensors. MIPs are indeed a promising tool to replace classical antibodies as recognition element. Working mechanisms of the 4 different kinds of sensing methods are well explained. It is also a good review of the use of nanomaterials to modify electrodes in different fields of application (clinical, food, environment), in relation to each of the 4 sensing method. Thanks for this interesting review. I have only a few comments or questions.

Neither figures nor tables are cited in the text. This would improve understanding if figures and tables were cited in the text.

Page 3, line 86:

Biolabeling instead of biolabling.

Page 5, line 139:

Nanoparticles instead of naonoparticles.

Page 5, line 148:

Replace “always” by “almost always” because there are some alternatives.

Page 10, line 354:

Sensorgrams instead of sensograms.

Page 13 Table 1:

Is it possible to indicate in the column “application” in which matrix the detection has been performed, as it is done for example for CYF? This column is not very useful if the matrix of application is not given because we have already the column template. For proof Table 2 for the electrochemical biosensors and table 3 for gravimetric biosensors do not have the application column.

The concentration units in the tables should be harmonised. Sometimes we have µM and sometimes µmol L-1 which is the same. We have also µg L-1 and µg/kg.

Reviewer 2 Report

The authors have written an excellent and comprehensive review of MIT for bioanalytical applications. It is a timely and important theme.

Few suggestions are provided for further improvement.

The title of the paper should be changed to "Advances in MIT for bioanalytical applications" as it would encompass all the elements of the current manuscript. The current title is quite verbose and misleading.

The advantages provided by MIT in comparison to alternative biorecognition elements such as antibodies should be mentioned clearly in a Table.

An overview of how the MIP is produced should be provided in more detail with figures in a separate section after the introduction.

The main advances in the synthesis of MIT should also be discussed also.

The challenges and pending concerns should also be discussed in a separate section at the end, which could be named as "Critiques and Outlook".

If there are some commercial companies or start ups looking into the MIT, these should be added to the text and summarized in a table.

Some interesting applications where the MIP have performed better than the conventional antibodies should be discussed to illustrate the potential of this technology.
